# Validation of the WHOQOL-BREF Quality of Life Questionnaire in an Urban Sample of Older Adults in a Neighbourhood in Zaragoza (Spain)

**DOI:** 10.3390/healthcare10112272

**Published:** 2022-11-12

**Authors:** Marta Gil-Lacruz, Miguel Cañete-Lairla, Jorge Navarro, Rosa Montaño-Espinoza, Iris Espinoza-Santander, Paulina Osorio-Parraguez

**Affiliations:** 1Grupo Bienestar y Capital (BYCS) Faculty of Education, Zaragoza University, C. Domingo Miral s/n, 50009 Zaragoza, Spain; 2Grupo Decisión Multicriterio Zaragoza (GDMZ), Departamento de Economía Aplicada, Facultad de Economía y Empresa, Universidad de Zaragoza, Gran Vía 2, 50003 Zaragoza, Spain; 3Department of Mathematics and Computer Science, Faculty of Science, University of Santiago, Santiago 9170022, Chile; 4Interuniversity Center for Healthy Aging, Faculty of Odontology, University of Chile, Santiago 8380544, Chile; 5Faculty of Social Sciences, Santiago de Chile University, Santiago 8370134, Chile

**Keywords:** health-related quality of life, neighbourhood, community surveys, questionnaires, older adults

## Abstract

Background: Nowadays, the increase in life expectancy needs to be matched by an increase in the wellbeing of older adults. A starting point is the definition of what is understood by health-related quality of life and its measurement in different contexts. Our research translates these international priorities to a local base. Objective: To evaluate the psychometric characteristics of the World Health Organization Quality of Life Questionnaire (WHOQOL-BREF) in a sample of older adults from a Spanish urban community (Casablanca). Methods: In collaboration with the local health centre, we designed and implemented the health neighbourhood survey. Interviews took place at subjects’ homes with 212 women and 135 men over the age of 60, who were residents in Casablanca. With the results, we evaluated the psychometric characteristics of WHOQOL-BREF and tested its reliability and validation. Results: The instrument has a high internal consistency with a Cronbach’s Alpha of 0.9. The items with higher correlation value were: ability to carry out activities in daily life, enough energy for daily life. The scale contributions of Physical Health dimension (0.809) and Psychological Health dimension (0.722) were notable. Conclusions: As with other studies, the instrument proved to be an integral evaluation of the diverse domains that condition the wellbeing of older adults.

## 1. Introduction

In 2016, the World Health Organisation [1] estimated that between 2015 and 2050, the number of people over the age of 60 will grow by 900 million and reach 2 billion: as a proportion of the world population, an increase from 12% to 22%. In countries such as Spain, this tendency is even more pronounced—the percentage of the population over 65 currently stands at 18.7%; by 2031 this figure will be 25.6% and by 2066 it will reach 34.6% [2]. This is a significant demographic shift that represents both an achievement and a challenge for society. An increase in life expectancy needs to be matched by an increase in the wellbeing of older adults [3]. 

Analysis of the keys to active and healthy ageing is therefore a social priority. This work must consider risk factors and morbidity as well as variables that can reduce vulnerability [4] and augment empowerment. This implies reviewing and looking for consensus regarding measurements of health and quality of life [5].

The need for the development of further research and knowledge in this area is recognised by public authorities; for example, the European Union Horizon Programme (2020) [6], with the aim of reducing social inequality, proposed that social agents (such as health systems) should make decisions based on data and the dissemination of good practices and innovative ideas. Such an approach could facilitate the design of public policies and programmes focused on positive ageing and educational strategies for the health of older adults [4,7,8]. This perspective argues for improved knowledge of the reality and needs of a heterogeneous collective (older adults) that often suffers from simplistic stereotyping. 

### WHOQOL-BREF Definition and Dimensions

A starting point is the definition of what is understood by health-related quality of life and its measurement in different contexts, along with suitable operationalisation criteria [9]. The WHOQOL-Group (1995:1403) defines Quality of Life as: “An individual’s perception of their position in life in the context of the culture and value systems in which they live and in relation to their goals, expectations, standards and concerns” [10].

In recent decades, the process of reaching this definition and the development of its corresponding measurement instrument has involved an international group of researchers and the participation of patients, families, care workers and health professionals [11]. 

The original questionnaire with 100 items was refined into the WHOQOL-BREF [12], which comprises 26 items and four domains of Quality of Life: Physical Health, Psychological Health, Social Relationships and Environment [13].

The reliability of the psychometric characteristics of the instrument was studied by the WHOQOL Group, using Chronbach’s Alpha for all the items of the scale (>0.7) and each of the domains (Physical Health: 0.82; Psychological Health 0.81; Social Relationships: 0.68; Environment: 0.80). A Multi-Trait/Multi-Item Analysis Program (MAP) gave positive results. The contribution of each item was significant when explaining the variance of the instrument as a whole [10,14].

Discriminant validity was studied by comparing the average scores obtained from samples of healthy and unhealthy subjects, utilising a t-test and regression analysis between the two generic items and the four domains of the questionnaire. The results of the multiple regression hierarchy showed that the joint impact of gender and age with regards to being healthy or unhealthy only explained 2.7% of the total variance (adjusted R²), although the equation is clearly significant (F_2,320_ = 96.3, *p* < 0.001). The correlations between the items and their corresponding domains oscillated between 0.48 for the evaluation of pain and 0.70 for daily life activities (Physical Health); from 0.50 negative feelings to 0.65 spirituality (Psychological Health); from 0.45 to 0.57 for sexual satisfaction (Social Relationships); and 0.47 for leisure to 0.56 for financial resources (Environment) [13]. The correlations for the Pearson test (a one-way test) and the domains were also high, positive and significant (*p* < 0.001): between 0.46 (Physical Health and Social Relationships) and 0.67 (Physical Health and Psychological Health). The exploratory analysis with varimax rotation demonstrated that the model with four domains is best adjusted to measure the Quality of Life in both healthy and unhealthy populations. The four domains (eigenvalues >1.0) explain 53% of the data variance [13]. An additional confirmatory factor analysis was undertaken to re-evaluate the adjustment of the domains to the model; the sample was divided into two random subsamples which gave a positive result (n = 5133 and n = 5872) [14]; the classification criteria were whether the interviewee felt healthy (n = 3313) or unhealthy (3862).

The instrument has also been validated with: older adults in a number of countries [15,16,17]; in Ibero-America [18,19]; with different health conditions [20,21,22]; and autogenous resources, including: Physical Activity [23] and Spirituality [24]. 

Recent studies have also focused on older adults, in correlation with other widely validated instruments: for example, Gobbens and van Assen (2014) [25] carried out a longitudinal study (2008, 2010 and 2012) with 484 subjects over the age of 75, with the aim of validating the WHOQOL-BREF and the Tilburg Frailty Indicator Questionnaire (TFI). Both instruments collect information relevant to an individual’s health and health complications (e.g., indicators of disability and use of health services). The results of this work showed that over time, the correlations between the domains of both scales were consistently high: in the WHOQOL-BREF, the values of the correlations varied between 0.41 and 0.73 in 2008; 0.52 and 0.77 in 2010; and 0.47 and 0.78 in 2012 (the strongest correlation was between Physical Health and Psychological Health). In the TFI, the result for Cronbach’s Alpha was also high (0.70).

The physical aspects of fragility in older adults (walking difficulties, balance, lack of strength in the hands and tiredness) are clearly associated with the Physical Health domain of the WHOQOL-BREF (average score 0.5). Three psychological aspects of fragility (feeling sad, nervous, anxious and inability to confront problems) also have a strong average correlation (0.37) with the WHOQOL-BREF. With regards to social fragility, only the absence of social support indicated a moderate correlation with the Social Relationships and Environment domains of the WHOQUOL-BREF.

The WHOQOL-BREF has been shown to be the most sensitive to the physical and psychological conditions of an individual. In a sample of people over 65 in Taiwan (n = 454), and after a multiple lineal regression analysis, the results of the four domains of the WHOQOL-BREF significantly covaried with results obtained from the depression scales of the Geriatric Depression Scale and the Modified Barthel Index [20]. 

From a complementary perspective, the instruments are also sensitive to salutogenic variables; for example, the sense of coherence in hospitalised senior citizens is a protector against depression (measured by the Hospital Anxiety and Depression Scale—HADS) and impacts the evaluation of the Physical Health, Psychological Health and Environment domains of the WHOQOL-BREF [26]. 

Due to the importance of suitable measurements of quality of life and health perception, the convergence of these instruments is of interest and leads to new lines of research such as the selection of a specific instrument for older adults. Costa, Driusso and Oishi (2014) [27] compared the WHOQOL-BREF with the Survey Health Status 36 (SF-36) using a sample of 278 older adults from Brazil; they found that while the psychometric properties of both methodologies were good (e.g., Cronbach’s Alpha was 0.832 for the former and 0.868 for the latter), the correlations between the domains of the questionnaires were not. The authors suggest that the SF36 should be used to measure the health and functionality of the patient and the WHOQOL-BREF as a more generic assessment focused on quality of life. 

There is a great deal of scientific literature on the cross-validation of the WHOQOL-BREF and the WHOQOL-OLD (which incorporates the measurement of specific domains: sensory abilities; autonomy; functional independence; social participation; death and dying; and intimacy) [28,29,30]. Both instruments are valuable for the measurement of health-related quality of life in clinical environments, for the evaluation of epidemiological services/programmes and primary care for older adults [31].

If we consider that health-related quality of life also depends on community factors such as environmental aspects and housing conditions, we should not overlook the importance of validating the instruments with local population samples [32]. The WHO has put forward measures to facilitate local questionnaires on this issue, including working with local groups and adapting previously utilised tools [33].

An important meta-analysis at a neighbourhood level [34] highlighted the influence of socioeconomic status on social relationships (e.g., measured through friendships and high-quality social interactions) on health-related quality of life. The dimensions combined objective elements (e.g., the number of social interactions) with personal perceptions and/or feelings (e.g., quality of relationships, friendships, support). 

The complementary nature of the domains of the model were clearly observed in a Columbian study in which senior citizens indicated that the presence of local parks and recreation areas in their neighbourhoods were important elements for their lifestyles, socialisation and wellbeing [35].

Of further interest is the validation of the WHOQOL-BREF with a sample population of the communes in Santiago de Chile [36]. The work found that the older adults were quite dependent on the neighbourhood as both a place of social relationships and a structure for healthy spaces. This finding has obvious implications for the design of strategies for the promotion of healthy ageing [37]. The psychometric characteristics of the instrument revealed high internal consistency, for both the overall total of the scale (α = 0.88) and the individual questions/items (between 0.87 and 0.88) [36].

The objective of this current work is to add to the previously mentioned studies by analysing the psychometric properties of the WHOQOL-BREF using a sample of older adults who live in the Casablanca neighbourhood of the city of Zaragoza in northern Spain. The information collected will permit inferences to be made and pertinent conclusions will be dawn on the possible development of strategies and actions that could be used with older adults in an urban local community context. 

## 2. Method

### 2.1. Subjects and Procedures

The study universe comprised people over the age of 60, resident in the neighbourhood of Casablanca, in the city of Zaragoza, Spain. Casablanca is characterised by its socioeconomic contrasts, an older population of rural origin and the territorial and demographic expansion that has taken place in the area in the last decade; it exemplifies the processes of urbanisation and speculation that have occurred in other parts of the city. The area is delimited by roads, canals and streets and there are three, very different, residential zones: Viñedo Viejo (the original centre of the neighbourhood); Las Nieves (a middle class area with high levels of unemployment); and Fuentes Claras (a modern, very expensive residential area populated by professional families with a high level of formal education).

Prior to the implementation of the survey, the research team distributed a letter to every household, giving information on the objectives of the project and encouraging participation. 

The questionnaire was completed by means of a personal interview in the home of the subject, a process that took, on average, 25 min. Consent was given by each participant and the study was given approval by the WHOQOL Centre in Barcelona.

According to the Aragon Institute of Statistics, in 2013, the population of the Casablanca Health area over the age of 15 was 8324; comprising 3959 men and 4365 women. Moreover, 843 men and 805 women were between the ages of 15 and 29; 192 men and 204 women were between the ages of 30 and 59; and 1195 men and 1556 women were over 60 years old.

Taking into account the proportion of men (0.4735) and women (0.5247), and with a confidence level of 95.45%, a margin of error or precision of ±3% and adding an approximate reserve of 6%, a representative sample of the study universe was calculated as 1,037. The number of subjects over the age of 60 was established as 342.

Formula for calculation of sample size (n)
(1)$$n=\frac{\left[N*Z^{2}*p*q\right]}{d^{2}*\left(N-1\right)+Z^{2}*p*q}\;N\;\left(\mathrm{total}\;\mathrm{population}\right)=8324$$

Z^2^ = 2^2^ (confidence level of 95.45)

*p* (proportion of men) = 0.4756

q (proportion of women) = 0.5244

d (precision or error) = 0.03 (3%)

The fieldwork was organised through a system of randomly selected routes and quotas that were stratified by sex and age. The routes were distributed in accordance with population density in the three areas of the neighbourhood. There were 10 routes with approximately 100 questionnaires for each route. Six routes were randomly selected for Viñedo Viejo; three for Las Nieves; and one for Fuentes Claras. As the model was polytypic by group, the first step was the random selection of a main entrance (in a block of flats); second, the selection of the floor (group of dwellings); third, the flat/residence (group of individuals); and finally the selection of the individual, based on sex and age (‘Young’: 16–29; ‘Adult’: 30–59; ‘Older Adult’: 60+) (sample unit). In order to achieve a representative sample of a minimum size necessary for the objectives of the study, a purposive sample was drawn in phases 2ª to 4ª (floor, flat, sex, age).

A pilot study with 35 participants tested the applicability of the characteristics of the scale and ensured that the questionnaire could be easily understood by the participants. Then, 347 older adults, resident in the neighbourhood were interviewed. There were 135 men and 212 women; a total of 263 were between the ages of 60 and 79 (third age) and 84 over the age of 80 (fourth age). The average age was 72, with a mean of 71 and a typical deviation of 9 years. The oldest participant was 97. 

The composition of the age groups and their frequencies in the sample universe (the neighbourhood) can be seen in Table 1.

As shown in Table 2, the majority of participants were married (66.6%), although 28% were widowed. More than half had primary education studies (51.9%), whilst 17.6% had no formal education and 18.2% had a school leaving certificate or vocational studies. These figures corresponded with the area of residence. A total of 66.6% lived in the old town, just 6.9% were from the new, luxury housing complexes and 23.6% were in the intermediate location. A total of 68.9% had sufficient economic resources each month, 13% stated that they had economic difficulties and 16.7% said they were able to save money each month.

A total of 75.5% of the respondents said that they had received a medical diagnosis.

### 2.2. Instrument

In addition to the socio-demographic and state of health questions, the instrument integrated the WHOQOL-BREF, which has 26 items: 2 general questions on quality of life and 24 others grouped into 4 domains—Physical Health, Psychological Health, Social Relationships and Environment. Each item was scored on a Likert-type 1–5 response scale, for example, ‘very poor’ (1) to ‘very good (5); ’not at all´(1) to ‘extremely’(5); ‘very dissatisfied’ (1) to ‘very satisfied’ (5); etc. Higher scores indicate a higher quality of life.

According to the WHOQOL-BREF instruction manual, the two initial items should not be included in a domain because they make reference to the generic evaluation of the individual on their quality of life and health [33]. These two independent points of reference have been shown to have a high correlation with the four domains [13]. 

Versions of the WHOQOL-BREF in different languages can be found on the World Health Organisation website: (https://www.who.int/tools/whoqol/whoqol-bref (accessed on November 10 2022).

### 2.3. Description of Data and Statistical Analysis

The reliability of the instrument was tested by means of Cronbach’s Alpha, as a measurement of the interrelationships between the items of the scale. Values higher than 0.7 are considered acceptable [38].

Validation of the instrument involved determining whether its application to the Casablanca sample distinguished the four domains of the WHOQOL-BREF. For this and following the instructions of the scale manual, items 3, 4 and 26 were recoded and these domains were constructed from the sum of the corresponding elements indicated in the manual, as long as there was, at most, a single missing value between the elements for each of the domains. As it is a Likert-type scale with a small number of categories, we define the variables Q3 to Q26 as ordinal, we calculate the Asymptotic Covariance Matrix that will be used as the weight matrix and will allow us to analyse the Correlations Moment Matrix.

The confirmatory factorial analysis examined the four domains (Physical Health, Mental Health, Social Relationships and Environment) with the aim of determining the contribution of each one to the concept of quality of life. The results of Mardia’s (1970) [39] multivariate normality were 5515.66 for skewness and 27.91 for kurtosis. It was therefore not possible to confirm compliance with multivariate normality in the items of the test, as a whole. The results of a Shapiro–Wilk test also failed to confirm multivariate normality for the individual items of the test. 

Given the previously mentioned results, it was necessary to use a robust technique for the confirmatory factorial analysis: the lack of compliance with multivariate normality was linked to the fact that the measurement scales for the items provide an ordinal scale with a reduced number of categories. For these reasons, it was decided to use Diagonal Weighted Least Squares as the estimation method for the model; the median and interquartile range were used as descriptive statistics as well.

SPSS 22.0, LISREL 8.80 and R, versions 3.33 and 3.5 were employed as the data treatment software.

## 3. Results

The first step was an analysis of lost data, it being useful to have information on the questions that were most left unanswered and questionnaires that were not completed with the minimum number of required responses. On average, between 2 and 11 questions were left unanswered; the four questions that were most frequently left unanswered were: *How satisfied are you with your sex life*? (48 unanswered); *How well are you able to get around*? (24 unanswered); *How much do you feel that pain prevents you doing what you need to do*? (22); and *How much do you need medical treatment to function in your daily life*? (20). 

Following the WHOQOL recommendations, questionnaires with more than 20% of the questions unanswered were discounted: 13 participants were subsequently eliminated, leaving a total of 334.

The frequency of the responses in the four domains of the WHOQOL-BREF was standardised to a range of 1 to 20 for each domain (Table 3). Minimum and maximum scores were obtained for each domain. The lowest score was for Social Relationships (median 13.3); the highest score was for satisfaction with Environment variables and Psychological Health (median 14.0). We found a greater variability in Physical Health and Social Relationships (interquartile range 4.0), while the lowest is Psychological Health (interquartile range 3.2). Skewness and kurtosis values are also indicated.

### 3.1. Factorial Structure

Lisrel 8.80 produced the information given in Figure 1, utilising Diagonal Weighted Least Squares. This method of analysis takes into account the metric quality of the variable, especially when the number of response categories for the items is relatively small [40]. In order to interpret the indices that are obtained, it is necessary to know the number of starting variables and the number of participants in the sample [41]. In this case, the adjustment indices were acceptable in consideration of the aforementioned criteria. Despite the fact that the value of ⎟^2^ that was obtained from the correction of the Satorra–Bentler (1994) scale [42] was very high, its coefficient of the degrees of freedom X^2^/df = 2.59 was acceptable. The values for the SRMR (=0.075), RMSEA (=0.070) and CFI (=0.96) are also indicative of an acceptable fit; both the GFI (=0.97) and AGFI (=0.96) indicate a good fit. The PGFI result was 0.79. Alternative models were considered—none of them achieved a valid solution with greater parsimony. Figure 1 shows the coefficients of the Completely Standardized solution for a better understanding of their meaning.

### 3.2. Internal Consistency

As can be seen in Table 4, Cronbach’s Alpha gave the reliability of the scale for the overall total of the items as 0.90 and the same was true for the discrimination indices for each individual item (corrected item–test correlation). In relation to the contribution of the items to the measurement of the instrument, it is worth underlining the high value of the question on ability to carry out activities in daily life, which had a correlation of 0.772. The next two questions with high scores were: *Do you have enough energy for daily life*? (0.724); and *How satisfied are you with your capacity for work*? (0.702). Just two questions had correlations lower than 3: *How satisfied are you with the support you get from friends*? (0.286); and *How satisfied are you with your access to health services*? (0.216).

The results for the internal consistency of the domains of the questionnaire were also acceptable (Table 5). The contributions of Physical Health (0.809) and Psychological Health (0.722) were especially notable.

The conceptual map of the questions obtained through the LISREL program and the application of Diagonal Weighted Least Squares gives the error for each of the items and the coefficient associated with the corresponding domain. The coefficients of the inter-domain correlations are also shown. 

## 4. Discussion

Despite the socioeconomic diversity of the participants, there were no problems of comprehension: only 13 respondents were removed due to failure to answer a sufficient number of questions. The percentage of unanswered questions (for example, How satisfied are you with your sex life?) was slightly higher than the 6% reported in a WHO study among the general population [13], but similar to other studies with older adults [12,43] and studies in Latin American contexts [36]. The explanation for the result could be found in the negative social representation of sexuality among these populations which may dissuade them from giving answers to more explicit questions [44]; another factor could be that being a widow/widower or not in a relationship might cause difficulties for the respondent [36].

Reliability was high for the instrument as a single measurement (a general scale) and in its four domains (Physical Health, Psychological Health, Social relations and Environment), these results coincide with the original WHO study [13] and similar studies with older adults [15,36]. 

The domain with the lowest level of reliability was Social Relationships. This might be due to the fact that there are only three questions in this section [36]. Good quality social relationships and a good social network are important factors in the wellbeing of older adults [34,37].

As with other research that has aimed to validate predicative instruments of health in older adults (e.g., prediction of fragility [25]), the results of the present study show that the scores for social relationships have the worst prognostic value. As previously mentioned, the small number of items/questions in this domain could be the reason for this finding [25].

We must emphasise that the most important protection for the elderly—counting with a modicum of social relationships—is often left out of scope in those health care systems, which are focused mostly on organ failures and age-related diseases [45]. One of the suggestions we could discuss, in light of the present results, is that via the coordination with municipal or regional day centers for isolated elders, the quality of life may be raised considerably and in some cases prevent anti-depressive medication and the harmful syndrome of polypharmacy in the elderly. This implies the development of integrative management plans for general health and the necessary interrelationship between medical care institutions and social care institutions. It is not only the sustainability of health care systems what is at stake, but the betterment of daily lives for a growing population of increasingly isolated elders in neighborhoods such as Casablanca [46].

Concretely in Casablanca, it could also be the case that the older population do not find it easy to evaluate their social relationships as their state of health might make such interactions more difficult. This possibility opens a new avenue for research into the sphere of relationships of this sector of the population. Both hypotheses—lack of satisfaction with social relationships due to isolation [47] and incapacity for social relations due to poor state of health [48]—obtained confirmatory results. Further research on these issues is necessary to determine their specific weights. 

The fact that the domains of health-related quality of life are highly correlated suggests that for older adults the concept is unidimensional when compared with its classic, multidimensional definition [10,14,49].

The confirmatory factorial analysis reflected the four domains of the instrument in its original version [13]. The maintenance of these factors supports the hypothesis that health-related quality of life has elements that are common to different cultures and sectors of populations (age, socioeconomic conditions, state of health, etc.) [36]. 

The main limitation of the present work would be the small sample size and the absence of longitudinal data that would allow temporal perspectives regarding the objective of the study. The adjustment indices of the model could be improved if they are applied to a general population as the concept of quality of life can vary among different age groups. 

Future lines of research that can be derived from this present work could be integrated into a community health questionnaire that would analyse the results from the perspective of gender, age groups and intergenerational groups, with greater emphasis on social stratification and interrelationships with other variables (health perception, consumption of medicines, sickness diagnoses, physical exercise, consumption of tobacco and alcohol, care of family members, community support, volunteering, perceived sources of health improvement, etc.).

## 5. Conclusions

From the results of this present study, it can be concluded that the WHOQOL-BREF is a reliable and valid instrument; the model proposed by the WHO measures health-related quality of life among community samples of older adults. As with other related studies, the instrument proved to be an integral evaluation of the diverse domains that condition the wellbeing of older adults and it offers valuable information on the daily experiences of people at this stage of their lives [9,50].

Although the use of the instrument can be clinical (e.g., for treatment options, quality of life of patients, etc.), the present study aimed to identify socio-sanitary needs in the community. The community aspect is important for the optimisation of efficiency and efficacy of the provision of health support and the reduction of inequalities by utilising data-based decisions. Our preliminary results suggest that the empowerment of older adults in their own social integration could be important for improving opportunities for healthy relationships. 

Our work has followed the WHO guidelines, with specific measures to facilitate local questionnaires on this important issue of the quality of life for the elderly, including working with local groups and adapting previously utilised tools. As we have clearly found, health-related quality of life crucially depends on community factors such as environmental aspects and housing conditions. 

The central objective of our work has been to add to all the previously mentioned studies the analysis, in a local population, of the psychometric properties of the WHOQOL-BREF using a sample of older adults who live in the Casablanca neighbourhood of the city of Zaragoza. The information collected has allowed us to make strategic inferences on the possible social development of the community and the planning of actions and policies that could be used with older adults in an urban local community context.

Nevertheless, there is an ample discrepancy between the much generalised findings and conclusions achieved in this work and the sample, which is very specific. So it cannot be generalised that the results from this tool are valid for any sample in any neighbourhood environment. It is valid and reliable for the sample that we have collected and it must be taken into consideration for other local studies along similar guidelines. We cannot forget that meaningful interpretation of the general out of the local is but one of the cornerstones of applied social sciences.

## Figures and Tables

**Figure 1 healthcare-10-02272-f001:**
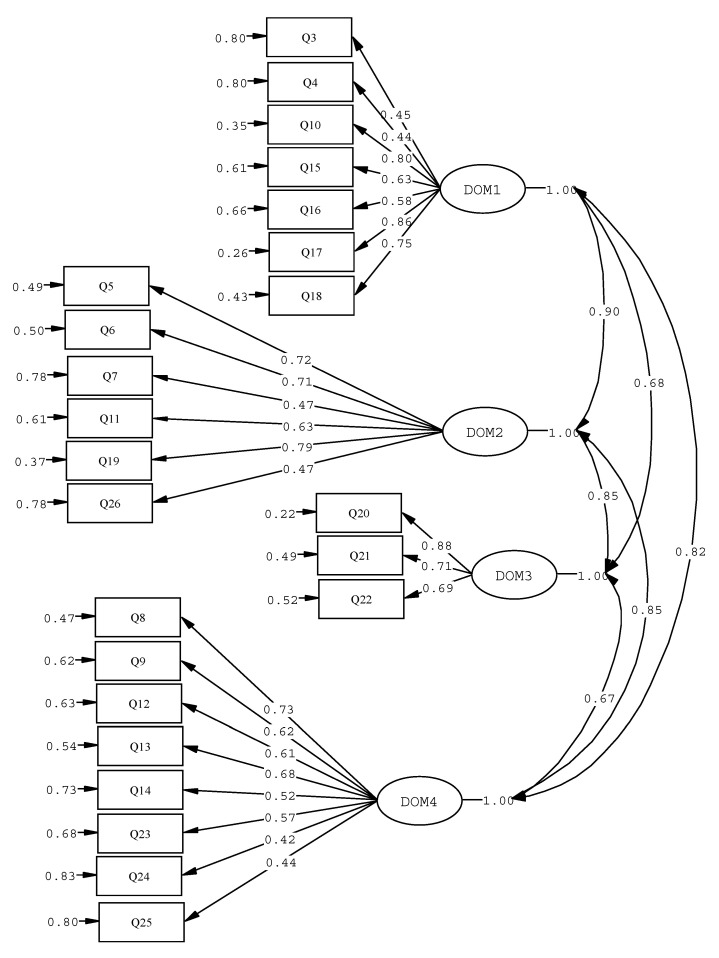
Confirmatory factorial model of the WHOQOL-BREF domains based on the 24 items of the scale, with Completely Standardized solution coefficients.

**Table 1 healthcare-10-02272-t001:** Composition of study universe and sample.

	Universe				Sample				
	Men		Women		Men		Women		
	Absolute Frequency	Relative Frequency	Absolute Frequency	Relative Frequency	No. of Tests	% Test	No. of Tests	% Test	Total
Third Age (60–79)	1000	45%	1180	55%	106	40.3%	157	59.7%	263
Fourth Age (80+)	195	34%	376	66%	29	34.5%	55	65.5%	84
Total	1195	43%	1556	57%	135	38.9%	212	61.1%	347

**Table 2 healthcare-10-02272-t002:** General characteristics of sample.

Variables	Absolute Frequency	Relative Frequency
*Civil status*		
Single	11	3.2
Married/cohabiting	231	66.6
Separated/divorced	6	1.7
Widow/Widower	97	28.0
NS/NA	2	0.5
*Education*		
No studies	61	17.6
Primary	180	51.9
Secondary	63	18.2
University	42	12.1
NS/NA	1	0.3
*Area of residence*		
Viñedo Viejo	231	66.6
Fuentes Claras	24	6.9
Las Nieves	82	23.6
NS/NA	10	2.9
*Perception of economic level*		
Problems of budgeting to end of month	45	13.0
Usually able to budget to end of month	120	34.6
No problem budgeting to end of month	119	34.3
Savings	58	16.7
NS/NA	5	1.4
*Medical diagnosis*		
Yes	262	75.5
No	64	18.4
NS/NA	21	6.1

**Table 3 healthcare-10-02272-t003:** Frequency analysis of scores obtained for the four domains of the WHOQOL-BREF.

Domain	Physical Health	Psychological Health	Social Relations	Environment
Description				
No. of items	7	6	3	8
No. of valid answers	322	322	322	322
Median	13.7	14.0	13.3	14.0
Interquartile range	4.0	3.2	4.0	3.5
Range	14.9	12.3	16	10
Skewness	−0.18	−0.26	−0.06	−0.04
Kurtosis	−0.26	0.02	0.32	−0.58
Minimum	5.1	6.7	4.0	9.5
Maximum	20.0	20.0	20.0	19.5

**Table 4 healthcare-10-02272-t004:** Results of item–test correlation and internal consistency.

Items	Corrected Item-Total Correlation
*Physical Health*	
Pain	0.450
Dependence on medication	0.431
Energy for daily life *	0.724
Mobility *	0.625
Sleep and rest	0.417
Daily life activities *	0.772
Capacity for work *	0.702
*Psychological Health*	
Positive feelings	0.559
Spirituality, religion	0.556
Thought, learning	0.322
Body image	0.449
Self-confidence *	0.677
Negative feelings	0.396
*Social Relationships*	
Personal relationships	0.511
Sexual activity	0.557
Social support **	0.283
*Environment*	
Freedom and safety *	0.603
Physical environment	0.434
Economic resources	0.458
Opportunity for information	0.578
Leisure and rest	0.457
Home	0.360
Healthcare **	0.216
Transport	0.380

* Questions with the highest item–test correlations—over 0.6; ** questions with the lowest item–test correlations—less than 0.3.

**Table 5 healthcare-10-02272-t005:** Results of the item–test correlation and internal consistency.

*Questionnaire Domains*	Internal Consistency (<)
Physical health	0.809
Psychological Health	0.722
Social Relationships	0.660
Environment	0.717
Total WHOQOL-BREF score	0.908

## Data Availability

The data presented in this study are available on reasonable request from the corresponding author. The data are not publicly available due to privacy restrictions.

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
