# Peer review of "Validation of the WHOQOL-BREF Quality of Life Questionnaire in an Urban Sample of Older Adults in a Neighbourhood in Zaragoza (Spain)"

_healthcare, 2022, doi:10.3390/healthcare10112272_

Round 1
Reviewer 1 Report
Dear authors,
I am glad I have this opportunity to review your manuscript entitled "Validation of the WHOQOL-BREF Quality of Life Questionnaire in an Urban Sample of Older Adults in Spain". Based on its reading my comments/notes are as follows:
Title
- it should be modified based on the parameters of your study (it is only a case study, so the title cannot be focused on the whole Spain population)
Abstract
- it should be rapidly rewritten
- highlight the importance of your topic
- focus on your own results (that are not mentioned at all)
- the last sentence is totally out
Introduction
- it represents two sections in one - you are not only introducing the topic but the majority of the text is also dedicated to one method - the WHOQOL-BREF questionnaire
Methods
- selection of respondents is described in detail, which I appreciate it
Results
- in the previous section, you mentioned that the Shapiro-Wilk test did not confirm normal distribution... so using the mean/average is useless
- description of the results is very short and brief (from my point of view, it has bigger potential)
Discussion
- no comments
I wish you all the best with this manuscript and other ones in the future.
Author Response
We would like to thank Reviewer 1 for the comments and solutions we have received. We have taken them into account one by one, and we are happy to share with you that not only the quality of the paper has improved considerably from the first version, but we have also learned a lot from this reviewing process.
Title
- It should be modified based on the parameters of your study (it is only a case study, so the title cannot be focused on the whole Spain population)
We have changed the title to this new version: Validation of the WHOQOL-BREF Quality of Life Questionnaire in an Urban Sample of Older Adults in a neighbourhood in Zaragoza (Spain)
Abstract
- it should be rapidly rewritten
- highlight the importance of your topic
- focus on your own results (that are not mentioned at all)
- the last sentence is totally out
Following the reviewers’ suggestions, our alternative abstract is:
Background: Nowadays, the increase in life expectancy needs to be matched by an increase in the wellbeing of older adults. A starting point is the definition of what is understood by health-related quality of life and its measurement in different contexts. Our research translates these international priorities to a local base.
Objective: To evaluate the psychometric characteristics of the World Health Organization Quality of Life Questionnaire (WHOQOL-BREF) in a sample of older adults from a Spanish urban community (Casablanca).
Methods: In collaboration with the local health centre, we designed and implemented the health neighbourhood survey. Interviews took place at subjects’ home with 212 women and 135 men over the age of 60, who were resident in Casablanca. With the results, we evaluated the psychometric characteristics of WHOQOL-BREF and tested its reliability and validation.
Results: The instrument has a high internal consistency with a Cronbach's alpha of 0.9. The items with higher correlation value were: ability to carry out activities in daily life, enough energy for daily life. The scale contributions of Physical health dimension (0.809) and Psychological health dimension (0.722) were notable.
Conclusions: As with other studies, the instrument proved to be an integral evaluation of the diverse domains that condition the wellbeing of older adults.
Introduction
- It represents two sections in one - you are not only introducing the topic but the majority of the text is also dedicated to one method - the WHOQOL-BREF questionnaire
As reviewer suggests we have add a new section entitle: WHOQOL-BREF definition and dimensions
Methods
- Selection of respondents is described in detail, which I appreciate it
Thanks for the comment; we tried to do our best in t
Results
- In the previous section, you mentioned that the Shapiro-Wilk test did not confirm normal .. so using the mean/average is useless
The mean and standard deviation have been removed from Table 3 and the non-parametric indices median and interquartile range have been added, with some comments. Symmetry and pointing measures by domains have also been included.
New version:
The lowest score was for the Social relationships (Median 13.3), the highest score was for satisfaction with Environment variables and psychological health (Median 14.0). We found a greater variability in Physical health and Social relationships (Interquartile range 4.0), while the lowest is psychological health (Interquartile range 3.2). Skewness and kurtosis values are also indicated.
Table 3: Frequency analysis of scores obtained for the four domains of the WHOQOL-BREF
|
Domain |
Physical Health |
Psychological Health |
Social Relations |
Environment |
|
Description |
|
|
|
|
|
No. of items |
7 |
6 |
3 |
8 |
|
No. of valid answers |
322 |
322 |
322 |
322 |
|
Median |
13.7 |
14.0 |
13.3 |
14.0 |
|
Interquartile range |
4.0 |
3.2 |
4.0 |
3.5 |
|
Range |
14.9 |
12.3 |
16 |
10 |
|
Skewness |
-0.18 |
-0.26 |
-0.06 |
-0.04 |
|
Kurtosis |
-0.26 |
0.02 |
0.32 |
-0.58 |
|
Minimum |
5.1 |
6.7 |
4.0 |
9.5 |
|
Maximum |
20.0 |
20.0 |
20.0 |
19.5 |
- Description of the results is very short and brief (from my point of view, it has bigger potential)
In descriptive results we have added a paragraph: The lowest score was for the Social relationships (Median 13.3), the highest score was for satisfaction with Environment variables and psychological health (Median 14.0). We found a greater variability in Physical health and Social relationships (Interquartile range 4.0), while the lowest is psychological health (Interquartile range 3.2). Skewness and kurtosis values are also indicated.
In discussion section, we have pointed a new argument about these results:
We must emphasize that the most important protection for the elderly –counting with a modicum of social relationships– is often left out of scope in current health care system, highly segmented and focused in organ failures and age-related diseases [45]. One of the suggestions we could discuss, under the light of the present results, is that via the coordination with municipal or regional day centers for the isolated elders, the quality of life may be raised considerably, and in some cases prevent anti-depressive medication and the harmful syndrome of polypharmacy in the elderly. This implies the development of integrative management plans for general health, and the necessary interrelationship between medical care institutions and social care institutions. It is not only the sustainability of health care systems what is at stake, but the betterment of daily lives for a growing population of increasingly isolated elders in neighborhoods such as Casablanca [46].
We have highlighted some of the main consequences related to the social relationships dimension, which we think provides a very fertile direction for future work. We have also entered some new general aspects of the discussion in the conclusions themselves, as these aspects were more general and richer in consequences.
Discussion
-No comments
I wish you all the best with this manuscript and other ones in the future.
We are very grateful for the positive feedback of Reviewer 1, because it gives us the strength to improve our work.

Reviewer 2 Report
Thank you for the opportunity to review this manuscript, which aims to analyze the psychometric characteristics of the World 17 Health Organization Quality of Life Questionnaire (WHOQOL-BREF) in a sample of older adults 18 from a Spanish urban community.
This is a validation of a questionnaire to measure quality of life. Despite being methodologically well structured, there are some considerations to take into account:
Abstract: structure by sections.
References: review journal instructions.
Introduction: justify the choice of this questionnaire to validate it, if there are other validated questionnaires with the same construct and same population. Also, reduce the introduction, it's too long and there's a lot of content that is expendable.
Material and methods:
- Numbers with "," must be included with 0 in front.
- How have the data of the characteristic of the sample of table 2 been obtained?
- Has the sample signed any type of informed consent?
- An explanation of the entire validation process is missing in the methodology section.
Results: Figure 1 has no quality, improve its sharpness.
Discussion: It would be better to put the discussion and conclusion sections separately.
The conclusion is very generalized, and the sample is very specific, it cannot be generalized that this tool is valid for any sample, it is valid and reliable for the sample that you have collected and it must be reflected in your conclusion.
After discussion all statuses, conflicts of interest, author contributions are missing....
Author Response
Thank you for the opportunity to review this manuscript, which aims to analyze the psychometric characteristics of the World Health Organization Quality of Life Questionnaire (WHOQOL-BREF) in a sample of older adults from a Spanish urban community.
This is a validation of a questionnaire to measure quality of life. Despite being methodologically well structured, there are some considerations to take into account:
We would like to thank the reviewer for the comments we have received. Our research has improved by their revisions. We have taken into account the suggestions that undoubtedly have helped to improve the quality of the paper.
Abstract:
Structure by sections
In the new version of the abstract, we have structure it by this way:
Background: Nowadays, the increase in life expectancy needs to be matched by an increase in the wellbeing of older adults. A starting point is the definition of what is understood by health-related quality of life and its measurement in different contexts. Our research translates these international priorities to a local base.
Objective: To evaluate the psychometric characteristics of the World Health Organization Quality of Life Questionnaire (WHOQOL-BREF) in a sample of older adults from a Spanish urban community (Casablanca).
Methods: In collaboration with the local health centre, we designed and implemented the health neighbourhood survey. Interviews took place at subjects’ home with 212 women and 135 men over the age of 60, who were resident in Casablanca. With the results, we evaluated the psychometric characteristics of WHOQOL-BREF and tested its reliability and validation.
Results: The instrument has a high internal consistency with a Cronbach's alpha of 0.9. The items with higher correlation value were: ability to carry out activities in daily life, enough energy for daily life. The scale contributions of Physical health dimension (0.809) and Psychological health dimension (0.722) were notable.
Conclusions: As with other studies, the instrument proved to be an integral evaluation of the diverse domains that condition the wellbeing of older adults.
References:
Review journal instructions
Thanks for this comment. We have changed all the references
Introduction:
Justify the choice of this questionnaire to validate it, if there are other validated questionnaires with the same construct and same population. Also, reduce the introduction, it's too long and there's a lot of content that is expendable.
Previous research has shown that WHOQOL-BREF is a useful test in order to measure a generic assessment on health quality of life. This measure represents a good alternative to analyze neighborhood contexts because includes social an environment dimensions. In addition, this study is part of a bigger research, which considers a broader population. Young and adults are going to be the next population to study. WHOQOL-BREF is an instrument, which allows us to analyze health quality of life through life span.
Following the reviewer suggestion, we have delate these paragraphs and their references:
In 2012, Brett et al. undertook a study that used the HADS and the International Personality Item Pool, and concluded that the results obtained with the WHOQOL-BREF influence both the characteristics of personality and depression in the Quality of Life of older adults.
In Spain, Lucas-Carrasco, Laidlaw and Power (2011) analysed the psychometric properties of the WHOQOL-BREF and the WHOQOL-OLD with a sample of 286 people over the age of 60, drawn from community centres, health centres, family associations and care homes. The internal consistency of the first was very high (.9); there was also good construct validity shown by results obtained with other scales such as the GDS-30 or the SF12. Results allowed the inference of statistically significant differences inside the collective (older adults) regarding educational level, physical and psychological health, availability of carers and living in a care home. The same research team studied the Attitudes to Ageing questionnaire and obtained similar results (Lucas-Carrasco, Laidlaw, Gómez-Benito and Power, 2013).
Brett, C.E., Gow, A.J., Corley, J., Pattie, A., Starr, J.M. y Deary, I.J. (2012). Psychosocial factors and health as determinants of quality of life in community-dwelling older adults. Quality of Life Research, 21 (3), 505-516.
Lucas-Carrasco, R., Laidlaw, K y Power, M.J. (2011). Suitability of the WHOQOL-BREF and WHOQOL-OLD for Spanish older adults. Ageing and Mental Health, 15 (5), 595-604.
Lucas-Carrasco, R., Laidlaw, K., Gómez-Benito, J. y Power, M.J. (2013). Reliability and validaty of the attitudes to ageing questionnaire (AQQ) in older people in Spain. International Psychogeriatrics, 25 (3), 490-499.
Material and methods:
- -Numbers with "," must be included with 0 in front
Thanks, the numbers were reviewed
- How have the data of the characteristic of the sample of table 2 been obtained?
As we tried to explain in the section Instrument, the neighborhood survey included socio-demographic and state of health questions. The text contains the following words: “Instrument: In addition to the socio-demographic and state of health questions, the instrument integrated the WHOQOL-BREF”
- Has the sample signed any type of informed consent?
Yes, consent was given by each participant (we sent a copy of this consent to the journal).
- An explanation of the entire validation process is missing in the methodology
A new paragraph devoted to explain the validation process has been writing:
Validation of the instrument involved determining if its application to the Casablanca sample distinguished the four domains of the WHOQOL-BREF. For this, and following the instructions of the scale manual, items 3, 4 and 26 were recoded, and these domains were constructed from the sum of the corresponding elements indicated in the manual, as long as there were, at most, a single missing value between the elements for each of the domains. As it is a Likert-type scale with a small number of categories, we define the variables Q3 to Q26 as ordinal, we calculate the Asymptotic Covariance Matrix that will be used as the weight matrix and will allow us to analyze the Correlations Moment Matrix.
- Results: Figure 1 has no quality, improve its
A new figure 1 has added, including a standardized analysis of the data (suggested by reviewer 1)
- Discussion: It would be better to put the discussion and conclusion sections
We have separated both
Conclusion
The conclusion is very generalized, and the sample is very specific, it cannot be generalized that this tool is valid for any sample, it is valid and reliable for the sample that you have collected and it must be reflected in your conclusion.
We thought that the last paragraph devoted to community explained this point. In order to be more precise, we have changed the first phrase for this one: In our study, the psychometric characteristics of WHOQOL-BREF are very positive. It is a reliable and valid instrument in our work with older adults residents of an urban neighbourhood. The model proposed by the WHO measures health related quality of life among community samples of older adults.
We have also added a full paragraph emphasizing this very aspect in the conclusions:
Our work has followed the WHO guidelines, with specific measures to facilitate local questionnaires on this important issue of the quality of life for the elderly, including working with local groups and adapting previously utilised tools. As we have clearly found, the health related quality of life crucially depends on community factors such as environmental aspects and housing conditions.
The central objective of our work has been to add to all the previously mentioned studies the analysis, in a local population, of the psychometric properties of the WHOQOL-BREF using a sample of older adults who live in the Casablanca neighbourhood of the city of Zaragoza. The information collected has allowed us to make strategic inferences on the possible social development of the community and the planning of actions and policies that could be used with older adults in an urban local community context.
Nevertheless, there is an ample discrepancy between the much generalized findings and conclusions achieved in this work, and the sample, which is very specific. So it cannot be generalized that the results from this tool are valid for any sample in any neighbourhood environment. It is valid and reliable for the sample that we have collected and it must be taken in consideration for other local studies along similar guidelines. We cannot forget that the meaningful interpretation of the general out from the local is but one of the cornerstones of applied social sciences.
After discussion all statuses, conflicts of interest, author contributions are missing....
Thanks for this suggestion. We have added this information:
Author Contributions: Conceptualization, MGL, IES, POP; field research and data curation, MGL; methodology, MCL, JNL, and RME, original draft preparation and writing, MGL, MCL, IES, review and editing MCL, JNL, POP, supervision and funding acquisition MGL, All authors have read and agreed to the published version of the manuscript.
Funding: This article received a grant from the Department of Science, University and Knowledge Society of the Government of Aragón (Spain) in charge of the reference research group of Well-being and social capital (BYCS) (ref. S16_20R, internal code 270–308).
Informed Consent Statement: Informed consent was obtained from all subjects involved in the study
Data Availability Statement: The data presented in this study are available on reasonable request from the corresponding author. The data are not publicly available due to privacy restrictions.
Conflicts of Interest: The authors declare no conflict of interest.

Round 2
Reviewer 2 Report
The authors have made all suggested changes and have improved the manuscript.